# Renormalization Analysis of Topic Models

**DOI:** 10.3390/e22050556

**Published:** 2020-05-16

**Authors:** Sergei Koltcov, Vera Ignatenko

**Affiliations:** Laboratory for Social and Cognitive Informatics, National Research University Higher School of Economics, 55/2 Sedova St., 192148 St. Petersburg, Russia; vignatenko@hse.ru

**Keywords:** topic modeling, renormalization theory, optimal number of topics, Renyi entropy

## Abstract

In practice, to build a machine learning model of big data, one needs to tune model parameters. The process of parameter tuning involves extremely time-consuming and computationally expensive grid search. However, the theory of statistical physics provides techniques allowing us to optimize this process. The paper shows that a function of the output of topic modeling demonstrates self-similar behavior under variation of the number of clusters. Such behavior allows using a renormalization technique. A combination of renormalization procedure with the Renyi entropy approach allows for quick searching of the optimal number of topics. In this paper, the renormalization procedure is developed for the probabilistic Latent Semantic Analysis (pLSA), and the Latent Dirichlet Allocation model with variational Expectation–Maximization algorithm (VLDA) and the Latent Dirichlet Allocation model with granulated Gibbs sampling procedure (GLDA). The experiments were conducted on two test datasets with a known number of topics in two different languages and on one unlabeled test dataset with an unknown number of topics. The paper shows that the renormalization procedure allows for finding an approximation of the optimal number of topics at least 30 times faster than the grid search without significant loss of quality.

## 1. Introduction

Topic modeling (TM) is a machine learning algorithm that allows for automatic extraction of topics from large text data. Nowadays, TM is widely used in different research fields such as social sciences [1], historical science [2], linguistics [3], literary studies [4], mass spectrometry [5], and image retrieval, among others [6]. However, to model a dataset, most of the topic models require the TM user to select the number of topics that, in practice, is an ambiguous and complex task. An incorrectly tuned topic model can generate both poorly interpretable and unstable topics or a set of topics that do not capture the overall topic diversity of data. In literature, the main approach to the selection of the number of topics is a sequential search [7,8] in a space of possible values with a certain step set by the user, which is done to maximize a quality metric as a function of the number of topics. Log-likelihood [9], perplexity [10], and semantic (topic) coherence [11] are some of the most widely used quality metrics in TM. However, maximization of these metrics based on sequential search is very time-consuming. Thus, there is an obvious need to optimize the process of selecting the number of topics. Luckily, the size of texts collections is often large enough for using methods from statistical physics. Thus, application of methods from thermodynamics for quality estimation of topic models recently proposed in [12,13,14] has allowed optimization of both hyperparameters and the number of topics. In works [12,13,15], it was demonstrated that Renyi entropy approach leads to the best results in terms of accuracy for the task of determining the optimal number of topics with respect to classical metrics such as log-likelihood, perplexity, and semantic coherence. However, Renyi entropy approach is also based on grid search and, therefore, is computationally expensive.

In this work, we propose a way to overcome this limitation, at least for some models. While testing the Renyi entropy approach, we found out that some functions of the output of TM (namely, density-of-states function [16] and partition function, which is defined in Section 2.3), which are used for Renyi entropy calculation, possess self-similar behavior. This finding led us to think about the possibility of using the renormalization technique when calculating Renyi entropy. While works [12,13,14] propose an application of non-extensive entropy to the task of topic model parameter selection including the number of topics and define how to calculate Renyi entropy for the output of topic models, our recent works [17,18] contain the first attempts to exploit renormalization to speed up the Renyi entropy approach. However, works [17,18] contain limited numerical results for only one topic model and lack a discussion on problems that have to be faced when defining renormalization procedure for topic solutions. Moreover, the behavior of the partition function is not considered in those works. The first and main goal of our work is to study the possibility of applying the renormalization theory to finding the optimal number of topics in flat probabilistic topic models based on the entropy approach developed in works [12,13]. The second goal of our work is to demonstrate the advantage of renormalization approach in computational speed for determining the number of topics, which is an extremely important task when working with big data. We demonstrate the applicability of the renormalization technique to the task of selecting the number of topics and describe the algorithm of renormalization for three topic models. Let us note that renormalization technique is used exclusively for fast approximation of Renyi entropy and allows us to avoid multiple time-consuming calculations; however, it can not serve as an inference algorithm of topic models.

Renormalization is a set of tools for simplification, or for coarse-graining of the system under consideration. A simple and illustrative example of renormalization in image retrieval is image compression. More precisely, renormalization consists of building a procedure for scaling the system, which preserves the behavior of the system. Theoretical foundations of the modern renormalization theory were laid by Kadanoff [19] and Wilson [20] and currently are widely used in percolation analysis and the analysis of phase transitions. Let us note that, to apply renormalization, the system should possess a property of self-similarity in order to be able to maintain its behavior under scaling transformation. Therefore, the application of renormalization is natural in fractal theory since fractal behavior is self-similar [21,22]. A classical example of renormalization in physics is its application to the models of Ising and Potts. To describe it, let us consider a two-dimensional lattice of atoms where each atom is characterized by its state. The number of states depends on a concrete task. For instance, in the Ising model, only two states are considered, while, in the Potts model, the number of possible states varies from 3 to 5 [23]. The procedure of renormalization groups the nearest nodes and replaces them with a new node according to some rule. Thus, in majority vote coarse-graining approach, the state of the new node is determined by the majority of the states of the group. This procedure is carried out over the whole lattice and results in a new configuration of atoms and can be performed several times. It is worth mentioning that successive coarse-graining leads to a rough approximation of the initial system and, therefore, to approximate results. However, renormalization is a successful technique that allows estimating critical exponent values [20] in phase transitions where other mathematical approaches are not applicable.

The rest of the paper is divided into the following sections. Section 2.1 introduces general assumptions of TM and briefly discusses parametric and nonparametric models. Section 2.2 reviews the earlier developed entropic approach [13] for selecting the number of topics. This approach is based on the application of non-extensive entropy and establishes a link between TM and statistical physics. Section 2.3 discusses self-similar behavior of the density-of-states function (to be defined further) of topic models. Section 3.1 adapts the renormalization procedure to the optimization of the number of topics in TM. Section 3.2, Section 3.3 and Section 3.4 describe algorithms of renormalization for three topic models: probabilistic latent semantic analysis (pLSA), latent Dirichlet allocation (VLDA) with variational Expectation-Maximization (E–M) algorithm, and LDA with Gibbs sampling inference (GLDA). Section 3.5 contains a description of the test datasets and model settings. Section 3.6, Section 3.7 and Section 3.8 contain the results of computer experiments for each model and compare the obtained results between the renormalization approach and the entropic approach. Section 3.9 describes an intuitive concept of selecting the number of topics for an unlabeled dataset along with illustrative numerical approbation of this concept. Section 3.10 reports the computational speed of the proposed renormalization approach comparing it to standard grid search methods, and demonstrates significant gain in time achieved by our approach.

## 2. Materials and Methods

### 2.1. Brief Overview of Topic Models

Before passing to our research, we would like to discuss some basic principles of TM. The TM approach assumes that a document collection has a finite number of latent topics. Each topic can be represented by a distribution of words. Every word has probabilities of appearing in each topic. In turn, topics are assigned to documents with different probabilities. Basic probabilistic topic models ignore the order of words in documents (‘bag-of-words’) and the order of documents in a collection and exploit a conditional independence assumption that document *d* and word *w* are independent conditioned on the latent topic *t* [24,25]. Therefore, the probability of word *w* in document *d* can be expressed as follows [24]:(1)$$p\left(w|d\right)=\sum_{t\in T}p\left(w|t\right)p\left(t|d\right)=\sum_{t\in T}\phi_{wt}\theta_{td},$$
where *t* is a topic, p(w|t) is the distribution of words by topics, and p(t|d) is the distribution of topics by documents. The results of TM are represented with two matrices, namely, matrix $\Phi :=\left\{\phi_{wt}\right\}\equiv \left\{p\left(w|t\right)\right\}$ which contains distribution of words by topics and matrix $\Theta :=\left\{\theta_{td}\right\}\equiv \left\{p\left(t|d\right)\right\}$ which contains distribution of topics by documents. Dimension of matrix Φ is W×T and dimension of matrix Θ is T×D, where *W* is the number of unique words in the document collection, *D* is the number of documents, and *T* is the number of topics. Since for most tasks, it is probabilistic models that are usually applied, here we focus on three most popular probabilistic models: a classical version of pLSA [24], a classical version of VLDA model [26] and GLDA model [27], which was proposed to increase the stability of TM. For a detailed description of the models, we refer the reader to Appendix A. Note that all these models share the problem of topic number selection.

An alternative to the above parametric topic models is the application of nonparametric methods or models. The main idea of nonparametric models is to choose a value of the model parameter (for example, the number of clusters in cluster analysis and the number of topics in TM) not through standard Bayesian methods of model selection, which require training a set of models and essentially conducting a directional search over the parameter grid, but to choose it by introducing a prior distribution on potentially infinite partitions of integers using some stochastic process that would give an advantage in the form of a higher prior probability for solutions with fewer clusters/topics. A classical example of such a process is a Chinese restaurant process [28] and Indian buffet process [29]. An important advantage of nonparametric methods is that even such complex stochastic processes allow rather simple and fast inference algorithms based on Gibbs sampling, which are very similar to Gibbs sampling algorithms discussed in [30]. Nonparametric variants of the LDA model that are based on hierarchical Dirichlet processes and the Chinese restaurant process are introduced and considered in works [31,32,33]. More complicated models that are based on the Indian buffet process are considered in [34,35]. Detailed surveys on nonparametric models can be found in [36,37] while a detailed introductory tutorial on nonparametric methods can be found in [38].

However, nonparametric models possess a set of parameters that significantly influence the results of TM. For instance, in work [33], the essential influence of hyperparameters on the output of hierarchical topic models was demonstrated. In addition, in work [15], it was shown that, in real applications, the HDP model does not allow for determining the number of topics in datasets with a known true number of topics. We do not include investigation of nonparametric models in this work due to the above drawbacks and we think that the study of such models deserves a separate paper. Thus, we will not give a detailed description of those models. The adaptation of the entropy approach for the analysis of nonparametric models, in turn, requires a separate investigation.

### 2.2. Entropic Approach for Determining the Optimal Number of Topics

An entropy-based approach proposed in [12,13] is based on a procedure of measuring the level of Renyi entropy of a topic model. Maximum entropy corresponds to the initial state of a topic model where the distributions of words and documents are either uniform (flat distribution) or random. Correspondingly, Renyi entropy of a trained topic model has a significantly smaller value. This difference in the values of the deformed entropy allowed for formulating a principle of searching for the optimal number of topics based on the search for the minimum Renyi entropy. Furthermore, as was shown in [13], the number of topics where the minimum of Renyi entropy is located coincides with the number of topics identified by human coders. This allows us to replace the optimization of topic model hyperparameters exploiting human markup with the search for the minimum Reni entropy with varying values of hyperparameters [15].

The calculation of the deformed Renyi entropy, where the deformation parameter q=1/T is inversely proportional to the number of topics, is based on a two-level model. Therefore, the range of obtained probabilities in matrix Φ is divided into two intervals thus splitting the vocabulary of a given dataset into two levels. The first level includes words with high probabilities ($\phi_{wt}>1/W$) and the seconds level includes words with low probabilities, correspondingly. Thus, one can define the density-of-states function as
(2)ρ=N/(WT),
where *N* is the number of words with high probabilities. The energy can be expressed as follows:(3)$$E=-\ln\left(\tilde{P}\right)=-\ln\left(\frac{1}{T}\sum_{w,t}(\phi_{wt}\cdot {𝟙}_{\left\{\phi_{wt}>1/W\right\}}\right),$$
where ${𝟙}_{\left\{\phi_{wt}>1/W\right\}}=1$ if $\phi_{wt}>1/W$ and zero otherwise. Thus, ln(ρ) is an analogue of the Gibbs–Shannon entropy (similar to [39]) and
(4)$$Z_{q}=e^{-qE+S}=\rho {\left(\tilde{P}\right)}^{q}$$
is the partition function of a topic solution [12]. According to the definition of Renyi entropy in Beck notation [40], we obtain that, for TM, the Renyi entropy can be expressed as follows:(5)$$S_{q}^{R}=\frac{\ln(Z_{q})}{q-1}=\frac{q\ln\left(q\tilde{P}\right)+q^{-1}\ln\left(\tilde{\rho }\right)}{q-1},$$
where q=1/T, *T* is the number of topics. It is remarkable that Renyi entropy has non-monotonous behavior with a clear global minimum. Such non-monotonous behavior is explained with the fact that Renyi entropy includes two divergent processes, namely, Gibbs–Shannon entropy decreases with the increase in the number of topics while the energy (Equation (Equation 3)) increases. Thus, there is a region where Gibbs–Shannon entropy is balanced by internal energy, and this region corresponds to the global minimum point of Renyi entropy.

### 2.3. Self-Similar Behavior of Topic Models

Passing to the renormalization technique, we would like to note that the mathematical formalism of Renyi entropy is successfully used to describe multifractal statistical systems [41,42]. Moreover, Renyi entropy is closely related to renormalization procedures [43]. However, the works on the investigation of fractal behavior in the models of soft clustering (TM, in particular) and, consequently, on renormalization procedures of such models are very limited [16,17,18].

In [16], the multifractal approach is applied to the analysis of the behavior of topic models. This work proposes to consider the results of TM as an embedding of the space of words into a lattice of size W×T, where *T* is the number of topics (the number of columns in matrix Φ), and *W* is the number of unique words (the number of rows in matrix Φ). Such an embedding is represented by the matrix Φ, where the size of each cell of the lattice $\epsilon =\frac{1}{WT}$. If the size of the vocabulary is fixed, then the size of cells is determined by the number of topics, moreover, if T→∞, then the size of cells tends to zero. Reference [16] investigates the behavior of the density-of-states function under the variation of the cell size through the ‘box counting’ algorithm. Fractal approach to the analysis of the results of TM allows for detecting areas of self-similarity of the word distribution density function when the number of topics is varied. Such regions are characterized by straight lines in bi-logarithmic coordinates. In addition, fractal analysis allows for determining the so-called transition region, which corresponds to the region of the minimum Renyi entropy, that is, the region containing the optimal number of topics [16]. However, search for such region results in the necessity to calculate a set of topic models with different numbers of topics which is an extremely computationally expensive procedure.

Since there are regions of self-similarity in functions of the output of TM, the renormalization technique can be used for finding linear and transition regions [18]. A drawback of work [18] is that only the density-of-states function was analyzed, therefore, the behavior of the sum of probabilities of words was not taken into account. However, here we aim to overcome this drawback by studying the behavior of the partition function under variation of the scaling parameter q=1/T. It will allow us to provide a more solid basis for the applicability of the renormalization approach since partition function includes both functions (the density-of-states and the sum of word probabilities) that are involved in the calculation of Renyi entropy.

In the classical renormalization procedure for two possible states of an atom, i.e., spin directions (↓↑), the process of summation of degrees of exponential functions is performed for the nearest neighbors (in the one-dimensional case) [44]. In the case of the two-dimensional grid, calculations are significantly more complicated since there are different ways to summation [44]. In the case of TM, theoretical inference of the procedure of summation of word distributions is extremely complicated since, first, the notion of the nearest neighbors is not defined, as in each new run of the same TM algorithm, topics are assigned random indexes. It follows that neighboring indexes do not mean anything—either topic similarity or dissimilarity. The latter thus has to be somehow estimated. Second, there are different approaches to estimating the similarity of topics which presents a separate problem. Third, the number of spin directions (the number of topics, $T=q^{-1}$) may vary from two to several thousand while renormalization procedures for physical systems deal with a small number of clusters. However, one can implement a renormalization procedure in TM through a calculation of the values of the partition function under the variation of scaling parameter $q=T^{-1}$. We calculate partition function (Equation (Equation 4)) for three topic models (pLSA, VLDA, GLDA) on two datasets and demonstrate that there are several linear regions of the partition function in bi-logarithmic coordinates. Since angles of inclination of the lines are different, it follows that coefficients of self-similarity are different in these regions. Therefore, one can expect that a topic model with a relatively large number of topics implicitly contains a set of models with lower numbers of topics, where all these models are proportional to each other. Correspondingly, one can organize a renormalization procedure (i.e., procedure of coarsening of a topic model), where one can obtain several topic solutions with a smaller number of topics from a topic solution with a larger number of topics. Due to the fact that the minimum Renyi entropy is located in the transition region, one can expect that we will be able to identify such a transition region when performing renormalization.

However, the above theoretical statements are based on the analysis of the behavior of the partition function as a function of the scaling parameter and should be tested in direct computer experiment. Moreover, when conducting experiments on renormalization, first, one has to take into account the particular algorithm of TM. Because the computation of the Φ matrix depends on the used algorithm, the mathematical formulation of the renormalization procedure should be algorithm-specific. Second, one may consider different criteria of similarity of topics that lead to several algorithms of renormalization. In this paper, we account for both of these factors.

## 3. Results

### 3.1. General Formulation of the Renormalization Approach in Topic Modeling

In general, the proposed renormalization procedure consists of sequential coarsening of a single topic solution and calculation of Renyi entropy at each iteration of coarsening. Basically, the procedure of coarsening consists of merging of topic pairs (pairs of columns from matrix Φ) into a new single topic (one column) and calculating the distribution of this new topic. In this paper, we investigate three approaches to choosing pairs of topics for merging:Selection of two most similar topics in terms of symmetric Kullback–Leibler (KL) divergence [45]: for topics $t_{1}$ and $t_{2}$, $KL\left(t_{1},t_{2}\right)=\frac{1}{2}\left(\sum_{w}\phi_{wt_{1}}\ln\left(\phi_{wt_{1}}\right)-\sum_{w}\phi_{wt_{1}}\ln\left(\phi_{wt_{2}}\right)\right)+\frac{1}{2}\left(\sum_{w}\phi_{wt_{2}}\ln\left(\phi_{wt_{2}}\right)-\sum_{w}\phi_{wt_{2}}\ln\left(\phi_{wt_{1}}\right)\right)$.Selection of two topics with the smallest values of local Renyi entropy. Here, local Renyi entropy is according to Equation (Equation 5), where only probabilities of words in that topic are considered.Selection of two random topics. In this procedure, two integer random numbers are generated in the range [1,T] that indicate the indexes of the chosen topics, and if they are not equal, then we merge these topics.

Below, we describe algorithms of renormalization for each of the three selected TM algorithms, accounting for their unique mathematical approaches to probability calculation.

### 3.2. Renormalization for the LDA Model with Variational E–M Algorithm

We consider the version of the LDA model proposed in [26] where the distribution of topics by documents (topic proportions) follows Dirichlet distribution with *T*-dimensional parameter α. As a result of such modeling, we obtain a matrix Φ and a vector of the hyperparameter α. The inference algorithm of the model is based on the variational E–M algorithm. A more detailed description of both can be found in [26]. The iterative calculation of the Φ matrix is based on the following formula [26]:(6)$$\mu_{wt}=\phi_{wt}\exp\left(\psi \left(\alpha_{t}+\frac{L}{T}\right)\right),$$
where *w* is the current word, *L* is the document length, ψ is a digamma function, and $\mu_{wt}$ is an auxiliary variable which is used for updating $\phi_{wt}$ during the variational E–M algorithm. We build our renormalization procedure exploiting this essential Equation (Equation 6) and obtain the following algorithm:We select a pair of topics $t_{1}$ and $t_{2}$ using one of the principles described in Section 3.1.We merge the topics. Based on Equation (Equation 6), we calculate the distribution of a new topic *t* resulted from merging of $t_{1}$ and $t_{2}$ as follows:
(7)$$\phi_{wt}:=\phi_{wt_{1}}\exp\left(\psi \left(\alpha_{t_{1}}\right)\right)+\phi_{wt_{2}}\exp\left(\psi \left(\alpha_{t_{2}}\right)\right).$$Furthermore, we should normalize the obtained values of $\phi_{\cdot t}$ so that it would satisfy $\sum_{w}\phi_{wt}=1$. Let us note that Equation (Equation 7) represents a linear combination of probability density functions (in particular, probability mass functions) of two topics, where the mixture weights are chosen to resemble in some sense an iteration step of the inference algorithm of the model. However, Equation (Equation 7) can not be considered directly as a mixture distribution since it does not sum up to 1. However, after normalization, we obtain, indeed, a probability distribution. Correspondingly, the values of vector α should also be recalculated. The hyperparameter of the newly formed topic *t* is assigned to $\alpha_{t}:=\alpha_{t_{1}}+\alpha_{t_{2}}$. Then, vector α is normalized so that $\sum_{t}\alpha_{t}=1$. At this step, columns $\phi_{\cdot t_{1}}$ and $\phi_{\cdot t_{2}}$ are dropped from matrix Φ and replaced with the single new column $\phi_{\cdot t}$. Therefore, the size of matrix Φ becomes equal to W×(T−1).We calculate the global Renyi entropy for the new topic solution (matrix Φ) according to Equation (Equation 5). The Renyi entropy calculated in this way is further referred to as global Renyi entropy since it accounts for distributions of all topics.

Steps 1–3 are iteratively repeated until there are only two topics left. Then, to study the behavior of the obtained global Renyi entropy and to find its global minimum, a curve of the entropy as a function of the number of topics is plotted.

### 3.3. Renormalization for the GLDA Model

This model is based on the classical model of LDA with Gibbs sampling [30], but, in contrast to the classical one, it assigns the same topic to a whole window of the nearest words [27], where the size of the window is selected by a user. Therefore, this model can be considered as a regularized version of LDA: just like in classical LDA, it has two hyperparameters of Dirichlet distributions, α and β, and, additionally, the size of the window that may be viewed as a regularizer. The model produces stable solutions, however, as it was found in a later work [13], it leads to distortion in the Renyi entropy resulting in a shift of its minimum away from that defined by the human mark-up. In the GLDA model, matrix Φ is estimated using the so-called granulated Gibbs sampling algorithm. First, counters $c_{wt}$ are calculated, where $c_{wt}$ is the number of times word *w* was assigned to topic *t*. Then, matrix Φ is calculated according to the following equation:(8)$$\phi_{wt}=\frac{c_{wt}+\beta }{(\sum_{w}c_{wt})+\beta W}.$$

We build the procedure of renormalization based on these counters and exploiting the relation (Equation 8) for calculation of the distribution for a newly formed topic. Thus, the algorithm of renormalization consists of the following steps:We select a pair of topics $t_{1}$ and $t_{2}$ using one of the principles described in Section 3.1.We merge the chosen topics. In terms of counters, the merging of topics corresponds to a simple summation of the counters. Therefore, the distribution of a new topic *t* resulted from merging of $t_{1}$ and $t_{2}$ can be calculated as follows:
(9)$$\phi_{wt}=\frac{c_{wt_{1}}+c_{wt_{2}}+\beta }{(\sum_{w}c_{wt_{1}}+c_{wt_{2}})+\beta W}.$$It is clear that the distribution of the new topic adds up to one. Note that, at this step, the number of columns in the Φ matrix decreases.We calculate the global Renyi entropy for the new topic solution (matrix Φ) according to Equation (Equation 5).

Steps 1–3 are iteratively repeated until there are only two topics left. Then, to estimate the optimal number of topics, we search for the minimum point of Renyi entropy among the values obtained at step 3.

### 3.4. Renormalization for the pLSA Model

The pLSA model is the simplest among the considered ones since it does not contain regularizers, and the only parameter of the model is the number of topics [24,25]. The algorithm of renormalization consists of the following steps:We select a pair of topics $t_{1}$ and $t_{2}$ using one of the principles described in Section 3.1.We merge the chosen topics. Due to the simplicity of this model and the absence of hyperparameters, the distribution of a new topic *t* resulted from merging of $t_{1}$ and $t_{2}$ can be calculated as follows:
(10)$$\phi_{wt}=\phi_{wt_{1}}+\phi_{wt_{2}}.$$Thus, the merging of the chosen topics corresponds to the summation of the probabilities of words under the selected topics. Then, we normalize the obtained column $\phi_{\cdot t}$ so that $\sum_{w}\phi_{wt}=1$ and replace columns $\phi_{\cdot t_{1}}$, $\phi_{\cdot t_{2}}$ with the single column $\phi_{\cdot t}$.We calculate the global Renyi entropy for the new topic solution (matrix Φ) according to Equation (Equation 5).

Steps 1–3 are iteratively repeated until there are only two topics left. Then, a curve of the obtained Renyi entropy as a function of the number of topics is plotted.

To assess the ability of the proposed renormalization procedure to determine the optimal number of topics, we first compare the behavior of Renyi entropy calculated based on ‘renormalized’ matrix Φ and Renyi entropy calculated based on successive TM with different numbers of topics. Second, we compare the location of the minimum point of the Renyi entropy calculated based on renormalization and the number of topics selected by humans. Third, we compare the accuracy of the approximations of the optimal number of topics obtained with the renormalization approach and with the sequential search. Below, we describe the datasets which were used for testing the renormalization approach and the results of numerical experiments.

### 3.5. Data and Computational Experiments

To evaluate the accuracy of our approach, we considered two datasets with the known number of topics. Moreover, we tested our approach on an unlabeled collection with unknown number of topics. Thus, the following datasets are considered:‘Lenta’ dataset (available at https://github.com/hse-scila/balanced-lenta-dataset): a set of 8624 news items in the Russian language from Lenta.ru online news agency. The documents of this dataset were assigned to one of ten categories (https://www.kaggle.com/yutkin/corpus-of-russian-news-articles-from-lenta). In total, the dataset contains 23,297 unique words.‘20 Newsgroups’ dataset (available at http://qwone.com/~jason/20Newsgroups/): a well-known set of 15,404 news items in the English language. The number of unique words in the dataset equals to 50,948. The documents of this dataset were assigned to one or more of 20 topic groups, but according to [46], this dataset can be described with 14–20 topics as some of them are in fact very similar.‘French dataset’: a set of 25,000 news items in the French language collected randomly from newspaper "Le Quotidien d’Oran" (http://www.lequotidien-oran.com/). The vocabulary of this dataset contains 18,749 unique words.

For each dataset, we performed TM employing three algorithms, namely, VLDA, GLDA and pLSA, in the range of 2–100 topics in the increments of one topic. The values of hyperparameters in GLDA were set as follows: α=0.1, β=0.1; and the window size was set to l=1 in the notations of work [27]. TM was conducted using the following software implementations: *BigARTM* package (http://bigartm.org) integrated into a package *TopicMiner* (https://linis.hse.ru/en/soft-linis) for pLSA; *TopicMiner* package for GLDA; *lda-c* package (https://github.com/blei-lab/lda-c) for VLDA.

Then, topic solutions on 100 topics for each model (VLDA, GLDA, pLSA) underwent renormalization. Source codes of renormalization for each of the three models are available here: https://github.com/hse-scila/renormalization-approach-topic-modeling. Based on the results of the renormalization, curves of Renyi entropy as functions of the number of topics were plotted. Next, the obtained curves were compared to the Renyi entropy curves plotted using successive TM. The minima obtained with all methods on both datasets are summarized in Table 1. A detailed discussion of the results reported in Table 1 is given in Section 3.6, Section 3.7, Section 3.8, Section 3.9 and Section 3.10.

### 3.6. Results for LDA with a Variational E–M Algorithm

First of all, we would like to demonstrate the self-similar behavior of the partition function (Figure 1). Lines represent linear approximations while dots represent real data, and the two colors represent the two datasets. One can observe several regions where the partition function in bi-logarithmic coordinates is similar to a linear function (with different coefficients in different regions). It follows that the partition function is self-similar in those regions and renormalization theory can be applied.

Figure 2 shows the Renyi entropy curve obtained by successive TM with the varying number of topics (black line) and Renyi entropy curves obtained by renormalization with the merging of randomly chosen topics for the Lenta dataset. Here, and further, minima are denoted by circles in the figures. The minima of ‘renormalized’ Renyi entropy fluctuate in the range of 8–24 topics. However, after averaging over five runs of renormalization, we obtain that the minimum coincides with the result obtained by successive calculation of topic models (Table 1) and is very close to the human mark-up.

Figure 3 demonstrates the renormalized Renyi entropy curves with randomly chosen topics for merging for the 20 Newsgroups dataset. The minima points of renormalized Renyi entropy for five runs lie in the range of 11–17 topics. Averaging over these five runs, we obtain that the minimum is very close to the minimum obtained by successive calculation and falls within the optimal range of topics.

Figure 4 demonstrates the renormalized Renyi entropy curve for both datasets where topics for merging are selected according to the minimum local Renyi entropy. Here, and further, the results for the 20 Newsgroups dataset are represented by solid lines and the results for the Lenta dataset are represented by dashed lines. For both datasets, the minima of renormalized Renyi entropy correspond to the ground truth and are very close to the results obtained without renormalization.

Figure 5 shows renormalized Renyi entropy curves for both datasets, where topics for merging are selected according to the minimum KL divergence calculated between each pair of topics. Figure 5 displays a significant distortion of the Renyi entropy curve obtained by means of renormalization. Thus, we conclude that renormalization based on minimum KL divergence is not applicable for the task of searching for the optimal number of topics.

### 3.7. Results for the GLDA Model

Figure 6 shows multi-fractal behavior of the partition function in certain regions for the two datasets. In the region of T∈[11,46] for the Lenta dataset and T∈[7,38] for the 20 Newsgroups dataset, one can observe large fluctuations that contradict self-similarity. We presume that this is a feature of a distorted or over-regularized model.

Let us note that the minimum of the original Renyi entropy obtained without renormalization is significantly shifted from the true number of topics for both datasets (Figure 7 and Figure 8). Therefore, we conclude that this model leads to distortions caused by its type of regularization. This echoes with work [15], where more types of regularization were studied, and where it was demonstrated that regularization can lead to distorted results. However, it is beyond the scope of this paper to study the influence of regularization on the Renyi entropy. We aim to test if the renormalization approach can identify the optimal number of topics for this model or if the minimum point is also shifted from the true number.

Figure 7 and Figure 8 demonstrate renormalized Renyi entropy curves for five runs of renormalization with randomly chosen topics for merging for both datasets. After averaging over these five runs, we obtain that the minima points of renormalized Renyi entropy are larger than the true values. However, for the Lenta dataset, the estimation obtained with renormalization is closer to the number of topics determined by human judgment than that obtained without renormalization.

Figure 9 demonstrates the renormalized Renyi entropy curves for both datasets, where topics for merging are selected according to the minimum local Renyi entropy. In general, when applied to GLDA, this type of renormalization leads to lower values of the entropy as compared to the sequential search approach. It also yields the number of topics larger than that determined by human judgment but closer to that than all the other considered methods.

Figure 10 shows renormalized Renyi entropies for the two datasets, where the topics for merging are selected according to the minimum KL divergence between them. In line with VLDA results, this figure demonstrates that such type of renormalization does not allow us to determine the optimal number of topics since the minima are not very pronounced and strongly shifted to the right.

### 3.8. Results for the pLSA Model

Figure 11 shows the multi-fractal behavior of the partition function in the framework of the pLSA model for both datasets.

Figure 12 and Figure 13 demonstrate five renormalized Renyi entropy curves corresponding to the five runs of renormalization with random merging of topics for the two datasets and the original Renyi entropy curves obtained with successive TM. After averaging over these five runs, we obtain that this type of renormalization provides quite good results which are close to the minima of the original Renyi entropy and to the number of topics determined by human judgement.

Figure 14 demonstrates the renormalized Renyi entropy curves for both datasets where topics for merging are selected according to the minimum local Renyi entropy. Renormalization of the pLSA model leads to lower values of Renyi entropy with respect to the original one; however, the shape and the location of minimum are almost similar. In line with VLDA results, this type of renormalization leads to the number of topics which is very close to the true number of topics.

Figure 15 shows renormalized Renyi entropy curves for both datasets where the topics for merging were selected according to the minimum KL divergence between them. However, one can see that the renormalized curve does not have a clear global minimum; therefore, this type of renormalization does not allow us to select the optimal number of topics.

As it was demonstrated above, the best type of renormalization in terms of accuracy corresponds to the renormalization with the minimum local entropy principle of merging. Thus, this type of renormalization will be applied for analysis of the third dataset.

### 3.9. A Concept of Selecting the Number of Topics for an Unlabeled Dataset

As it was demonstrated above in our work and in works [12,13], Renyi entropy can be applied for searching the optimal number of topics for different datasets. Moreover, the renormalization procedure allows us to significantly speed up this search. However, the location of minimum Renyi entropy may significantly depend on the type of topic model, i.e., on the type of regularization used in the model [15], which causes difficulties when searching for the number of topics for unmarked datasets leading to the problem of choosing a topic model. In this subsection, we would like to demonstrate the influence of model type on the results of Renyi entropy approach and show how the renormalization procedure can be applied for quickly selecting the number of topics.

We considered an unlabeled dataset in the French language as a test dataset. The following models are applied to this dataset: pLSA, VLDA, GLDA and, additionally, LDA with Gibbs sampling, which is considered as an auxiliary model and is used for finding Renyi entropy minimum by successive TM with the varying number of topics. Renormalization of LDA model with Gibbs sampling is discussed in detail in our work [18].

Figure 16 demonstrates Renyi entropy curves obtained by successive TM with the varying number of topics. One can see that behavior of Renyi entropy for pLSA and LDA with Gibbs sampling is almost identical and the minimum is located in the region of 16–18 topics. However, Renyi entropy for VLDA has a global minimum for nine topics. In turn, Renyi entropy for the GLDA model does not possess a clearly visible global minimum. As it was discussed above, the GLDA model may be unsuitable for TM [13] in general. Thus, based on comparison of three other models, we conclude that the optimal number of topics for the French dataset is about 16 topics. In Section 3.6, Section 3.7 and Section 3.8, we showed that the best approximation of the optimal number of topics is achieved by menas of renormalization with the minimum local entropy principle of merging. Thus, we demonstrate the results only of this type of renormalization (Figure 17). Renormalization curves of Renyi entropy demonstrate that the minimum corresponds to 14–18 topics. Moreover, the renormalization curves for all the models have almost identical behavior with the varying number of topics. However, the rate of calculation of renormalization curves is many times higher than the calculation of Renyi entropy by successive TM.

Hence, when dealing with a new unlabeled dataset, it is enough to conduct TM for 3–4 different topic models with a fixed large enough number of topics and then to implement renormalization procedure of the obtained topic solutions. Furthermore, based on the obtained renormalization curves, one needs to find the common area of topics where the minimum values of entropy are found. This sequence of actions allows us to avoid problems related to the choice of model type and the effect of regularization on the results of TM.

### 3.10. Computational Speed

Table 2 demonstrates the time costs of Renyi entropy calculations for T∈[2,100] performed using different methods. The third column reports the time required for successive runs of TM for T∈[2,100] in the increments of one topic, while the fourth column reports the time required for calculation of a single topic solution on 100 topics. Columns 5–7 demonstrate time costs of renormalization of a single topic solution on 100 topics with the three described above approaches to merging topics. One can see that renormalization provides a significant gain in time for all considered models which is essential when dealing with big data. In our case, the renormalization allows for reducing the time of calculations at least by 80%.

Our calculations demonstrate that the fastest procedures are renormalization with the random merging of topics and with the minimum local entropy principle of merging. The latter type of renormalization also produces the curve the most similar to that obtained from successive TM and provides the best estimation of the optimal number of topics in terms of accuracy. Merging of random topics leads to significant fluctuations in the location of the global minima of Renyi entropy, however, averaging over several runs allows us to approach both the human-determined optimum and the sequential search result, with a negligible increase in the time of calculation. Renormalization with the minimum KL divergence leads to the significant shift of the minimum point of Renyi entropy from the value obtained both with the sequential search and human mark-up, and, therefore, is inappropriate for our task. We conclude that the most convenient procedure in terms of computational speed and accuracy is the renormalization with the local minimum entropy principle of merging.

## 4. Discussion

In this work, we have proposed a renormalization procedure for determining the range of the optimal number of topics in TM and tested it with three topic models. Renormalization involves a procedure of merging pairs of topics from a solution obtained with an excessive *T*. The principle of selection of topics for merge has turned out to significantly affect the final results. We considered three criteria for selecting the topics for merging, namely, topics with minimum KL divergence, topics with the lowest local Renyi entropy and random topics. We have demonstrated that the best result in terms of computational speed and accuracy for all three topic models corresponds to the renormalization procedure with the merging of the topics with the minimum local Renyi entropy. In this case, our renormalization approach allowed us to speed up the calculations at least by 96% which corresponds to the gain in time equal to six hours for the Lenta dataset, 11 h for the 20 Newsgroups dataset, and 34 h for the French dataset, on average. It is worth mentioning that we tested our approach on relatively small datasets (8624, 15,404, and 25,000 documents), correspondingly, the gain in time could be a week or more when applying our approach to larger datasets. The KL-based approach does not allow us to determine the optimal number of topics since the curve of renormalized Renyi entropy is either monotonously decreasing or has a minimum, which is significantly shifted with respect to the minimum of the original Renyi entropy. The reasons why merging of similar topics according to KL divergence leads to the worst results are not yet clear and require further research. The approach based on the selection of random topics has significant fluctuations in the location of the minimum; therefore, one should run this type of renormalization several times and average the results. On average, the estimation obtained with this type of renormalization is as accurate as the estimation obtained with a sequential search.

Summarizing our numerical results, we conclude that the renormalization approach allows for effectively finding the region of the optimal number of topics in large text collections without conducting a complete grid search of topic models. However, our approach had certain limitations. First, as it was demonstrated in the numerical experiments, the renormalization approach allows us to find the approximation of the optimal number of topics only for those models where the Renyi entropy approach in general can be successfully applied for this purpose. Therefore, for over-regularized or improperly tuned models, neither sequential search Renyi entropy approach nor its renormalized version are able to detect the true number of topics. Second, for the considered topic models, the probabilities of words in topics depend on the number of documents containing these words. This means that, if a topic is well-pronounced, but represented in a small number of documents, its vocabulary will not be able to acquire probabilities large enough to form a separate topic and thus will be absorbed by other topics. Thus, topic models can detect topics that are represented in many documents and poorly identify topics with a small number of documents. Therefore, the Renyi entropy approach and, consequently, the renormalization approach allow for determining the number of large topics only. Third, in our work, the renormalization approach was tested only for two European languages and on relatively small corpora. Correspondingly, our research should be extended and tested on non-European languages and larger corpora. Fourth, we developed and tested the renormalization procedure only for three topic models; however, there are other topic models to which a renormalization procedure could also be applied. Fifth, we applied the renormalization technique only for finding the optimal number of topics and did not consider other hyperparameters of topic models which should also be tuned. Correspondingly, our research can be extended for the fast tuning of other topic model parameters which is a promising direction for future research.

## Figures and Tables

**Figure 1 entropy-22-00556-f001:**
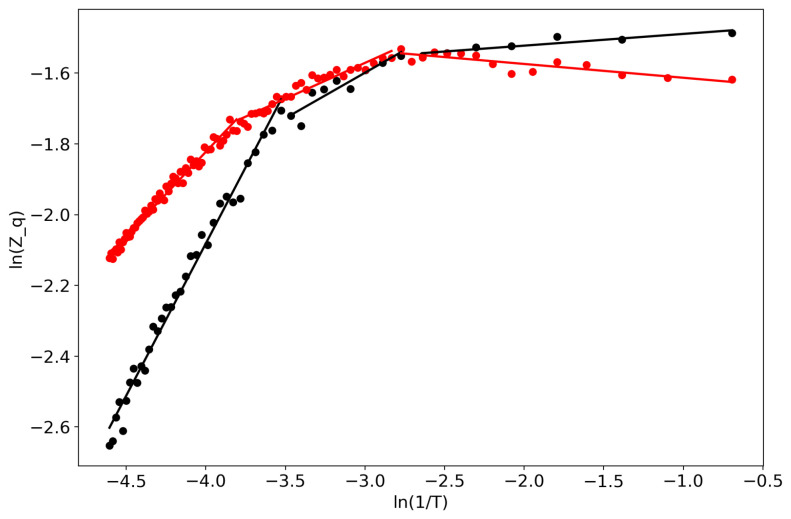
Partition function in bi-logarithmic coordinates (VLDA). Black: Lenta dataset; Red: 20 Newsgroups dataset.

**Figure 2 entropy-22-00556-f002:**
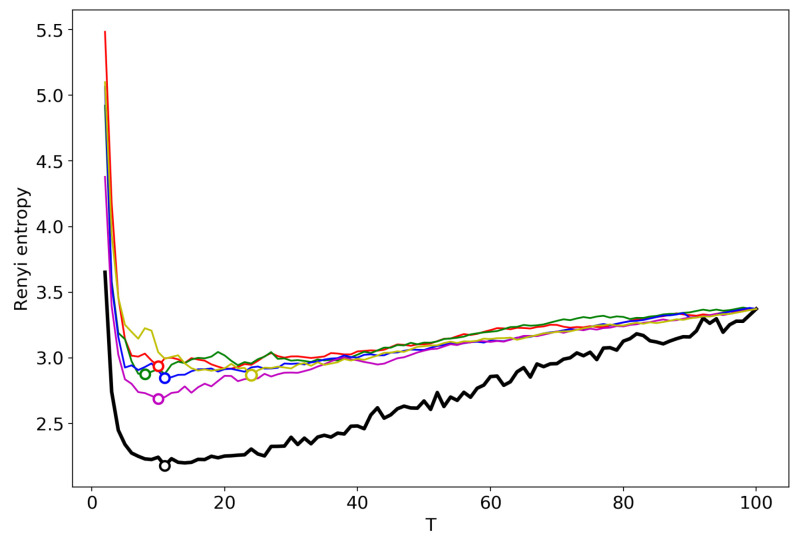
Renyi entropy curves (VLDA). Black: successive TM; Other colors: renormalization with randomly selected topics for merging; Lenta dataset.

**Figure 3 entropy-22-00556-f003:**
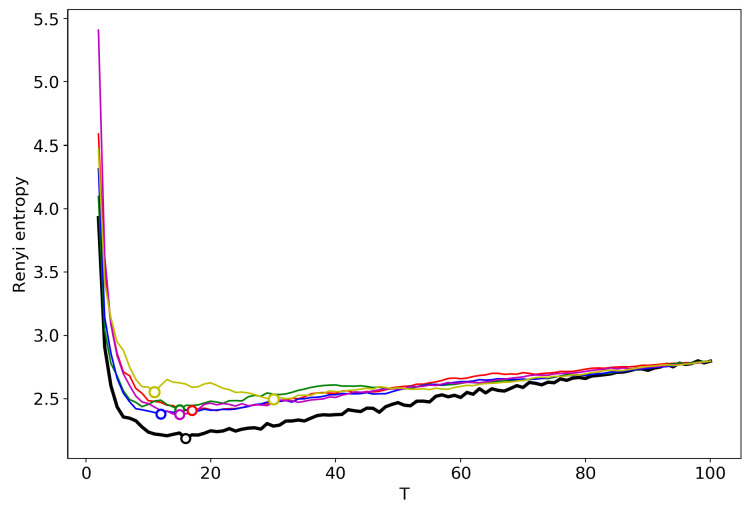
Renyi entropy curves (VLDA); Black: successive TM; Other colors: renormalization with randomly selected topics for merging; 20 Newsgroups dataset.

**Figure 4 entropy-22-00556-f004:**
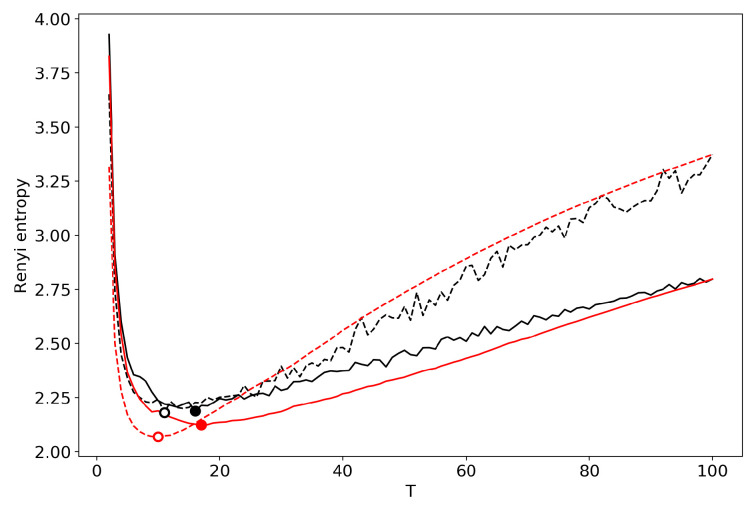
Renyi entropy curves (VLDA) for both datasets. Black: successive TM. Red: renormalization with the minimum local entropy principle of merging. Solid: 20 Newsgroups dataset; Dashed: Lenta dataset.

**Figure 5 entropy-22-00556-f005:**
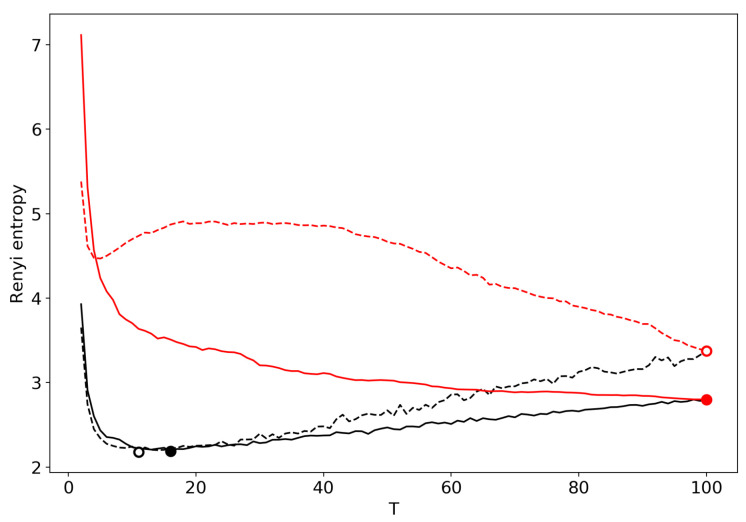
Renyi entropy curves (VLDA). Black: successive TM; Red: renormalization with the minimum KL divergence principle of merging; Solid: 20 Newsgroups dataset; Dashed: Lenta dataset.

**Figure 6 entropy-22-00556-f006:**
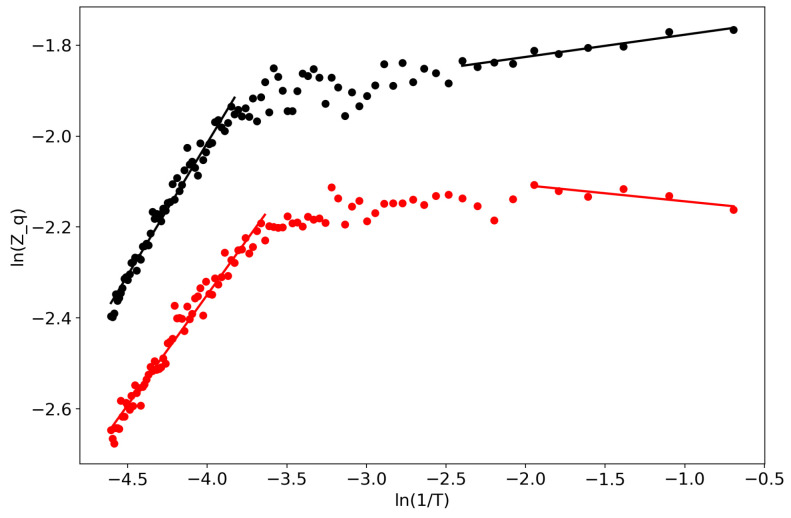
Partition function in bi-logarithmic coordinates (GLDA); Black: Lenta dataset; Red: 20 Newsgroups dataset.

**Figure 7 entropy-22-00556-f007:**
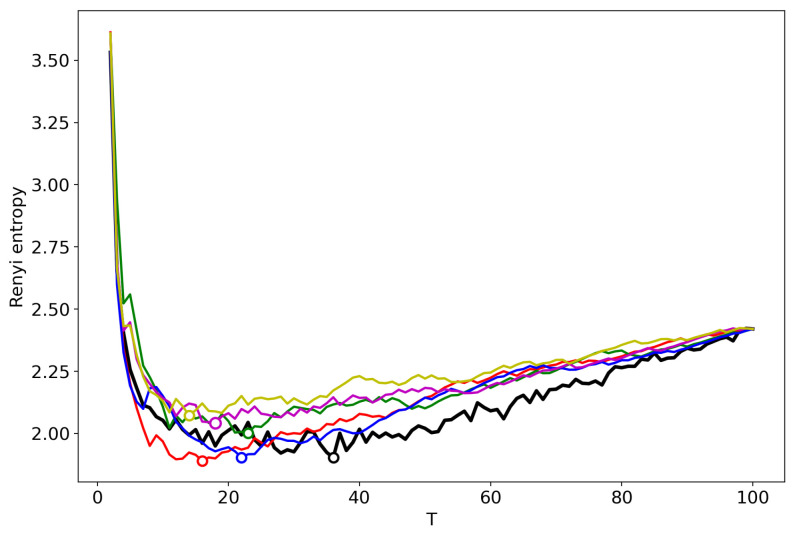
Renyi entropy curves (GLDA). Black: successive TM; Other colors: renormalization with randomly chosen topics for merging; Lenta dataset.

**Figure 8 entropy-22-00556-f008:**
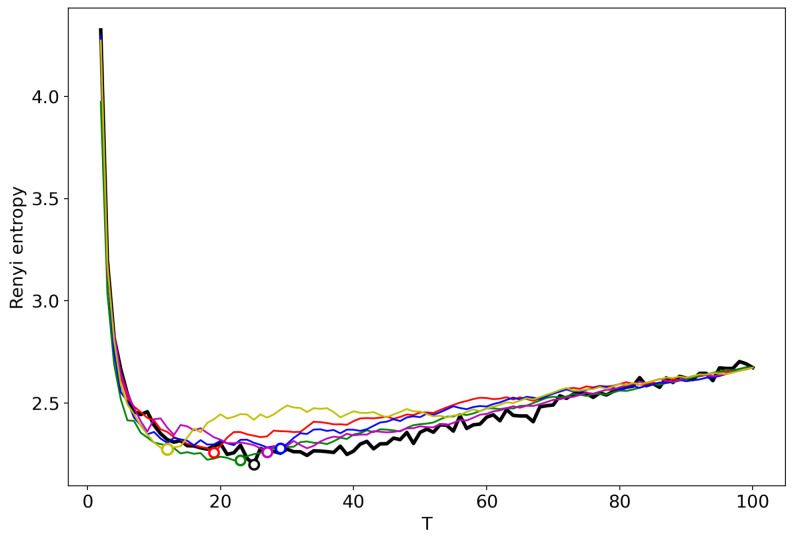
Renyi entropy curves (GLDA). Black: successive TM; Other colors: renormalization with randomly chosen topics for merging; 20 Newsgroups dataset.

**Figure 9 entropy-22-00556-f009:**
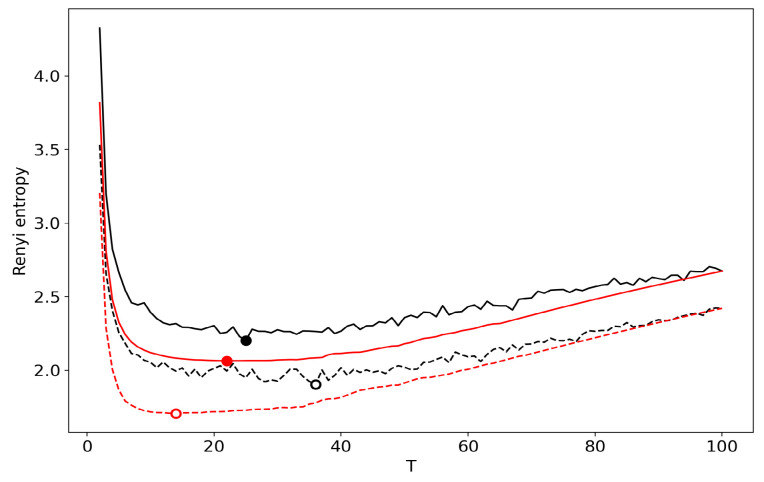
Renyi entropy curves (GLDA). Black: successive TM; Red: renormalization with the minimum local entropy principle of merging; Solid: 20 Newsgroups dataset; Dashed: Lenta dataset.

**Figure 10 entropy-22-00556-f010:**
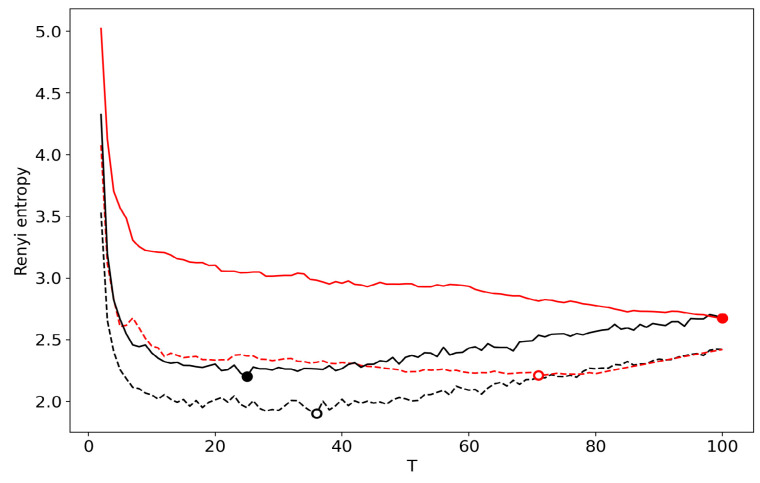
Renyi entropy curves (GLDA). Black: successive TM; Red: renormalization with the minimum KL divergence principle of merging; Solid: 20 Newsgroups dataset; Dashed: Lenta dataset.

**Figure 11 entropy-22-00556-f011:**
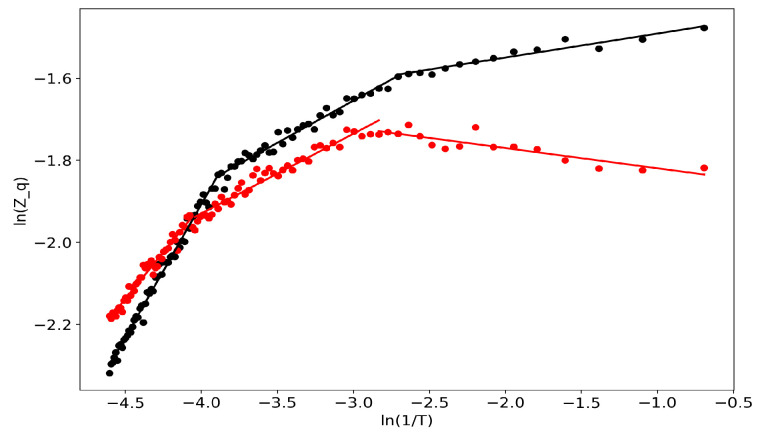
Partition function in bi-logarithmic coordinates (pLSA). Black: Lenta dataset; Red: 20 Newsgroups dataset.

**Figure 12 entropy-22-00556-f012:**
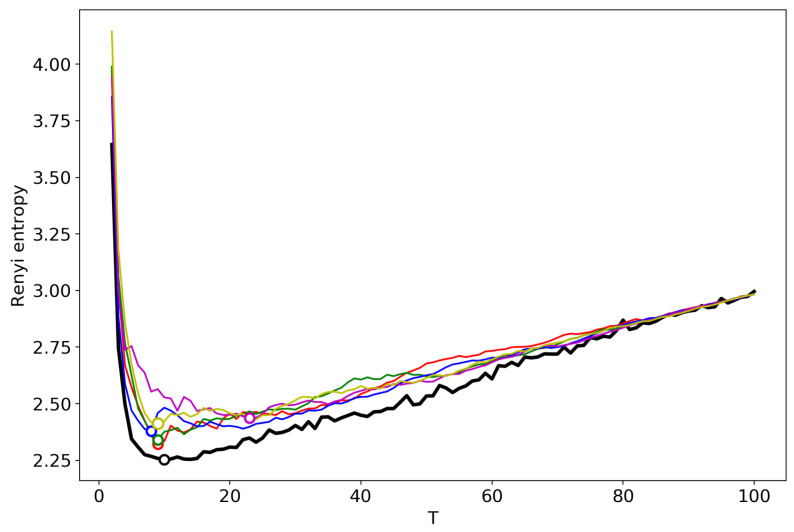
Renyi entropy curves (pLSA). Black: successive TM; Other colors: renormalization with the random merging of topics; Lenta dataset.

**Figure 13 entropy-22-00556-f013:**
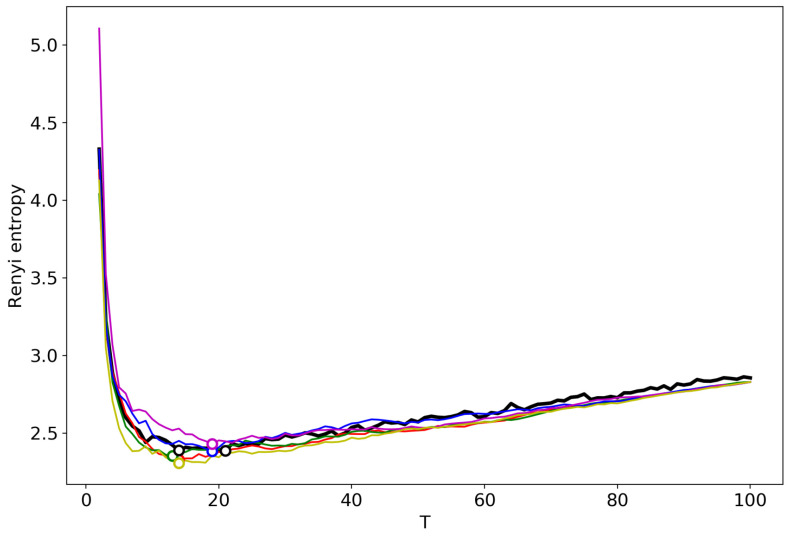
Renyi entropy curves (pLSA). Black: successive TM; Other colors: renormalization with the random merging of topics; 20 Newsgroups dataset.

**Figure 14 entropy-22-00556-f014:**
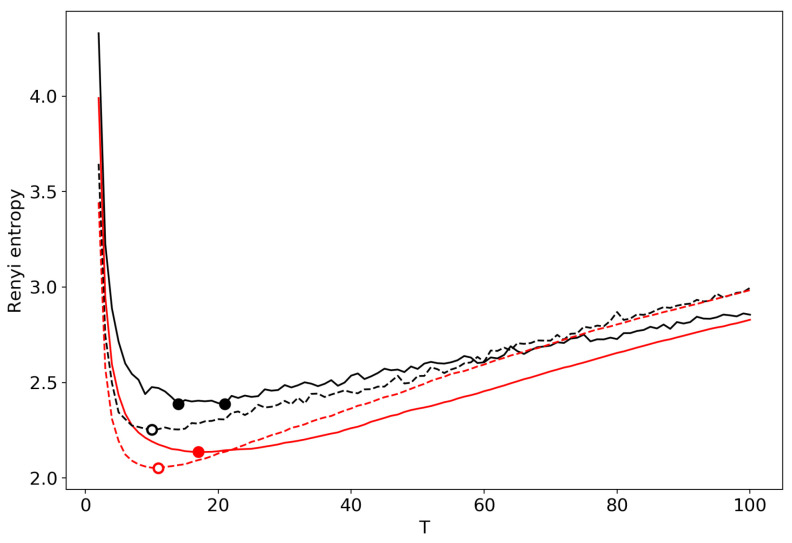
Renyi entropy curves (pLSA). Black: successive TM; Red: renormalization with the minimum local entropy principle of merging; Solid: 20 Newsgroups dataset; Dashed: Lenta dataset.

**Figure 15 entropy-22-00556-f015:**
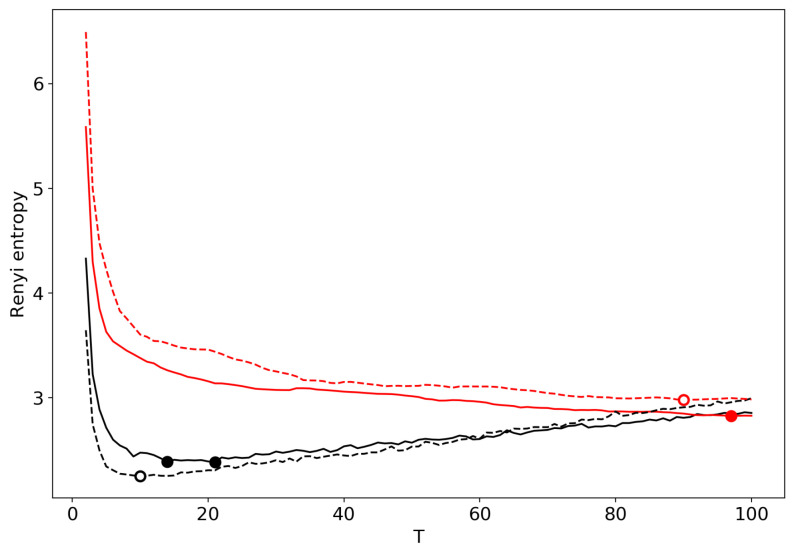
Renyi entropy curves (pLSA). Renyi entropy curves (pLSA); Black: successive TM; Red: renormalization with the minimum KL divergence principle of merging; Solid: 20 Newsgroups dataset; Dashed: Lenta dataset.

**Figure 16 entropy-22-00556-f016:**
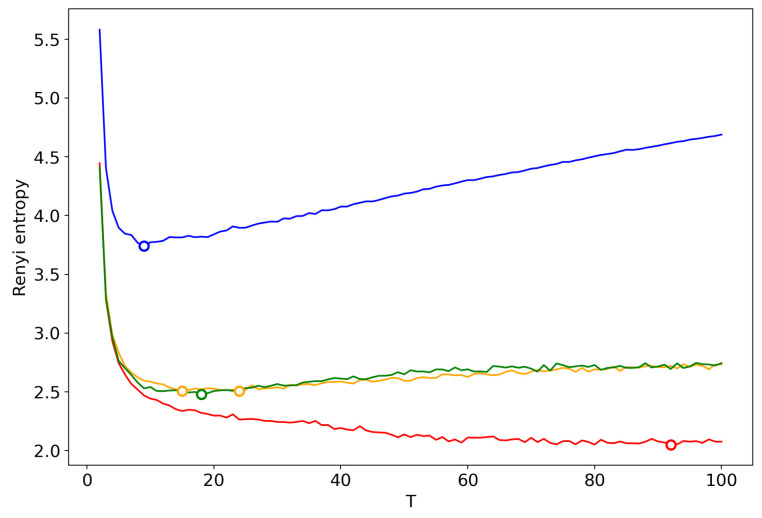
Renyi entropy curves (successsive TM). Blue: VLDA; Orange: pLSA; Red: GLDA; Green: LDA with Gibbs sampling.

**Figure 17 entropy-22-00556-f017:**
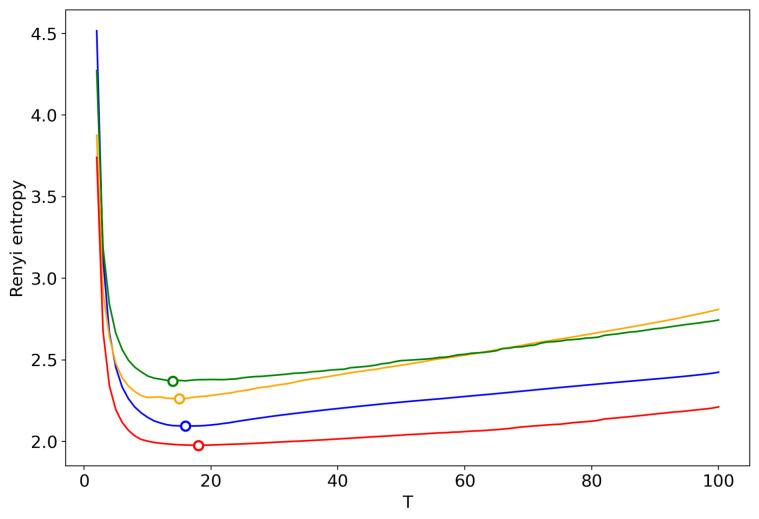
Renyi entropy curves (renormalization with the minimum local entropy principle of merging). Blue: VLDA; Orange: pLSA; Red: GLDA; Green: LDA with Gibbs sampling.

**Table 1 entropy-22-00556-t001:** Minima points of Renyi entropy obtained with different methods.

Dataset	T Search Method	Algorithm		
		VLDA	GLDA	pLSA
Lenta (10 topics)	Successive TM simulations	11	36	10
	Renormalization (random)	11	18	8
	Renormalization (min. Renyi entropy)	10	14	11
	Renormalization (min. KL divergence)	100	71	90
20 Newsgoups (14–20 topics)	Successive TM simulations	16	25	14
	Renormalization (random)	15	25	18
	Renormalization (min. Renyi entropy)	17	22	17
	Renormalization (min. KL divergence)	100	100	97
French dataset	Successive TM simulations	9	92	15; 24
	Renormalization (random)	93	40	25
	Renormalization (min. Renyi entropy)	16	18	15
	Renormalization (min. KL divergence)	100	100	99

**Table 2 entropy-22-00556-t002:** Computational speed.

Algorithm	Datase	Successive TM Simulations	Solution on 100 Topics	Renorm. (random)	Renorm. (min. Renyi Entropy)	Renorm. (min. KL Divergence)
pLSA	Lenta	360 min	9.2 min	0.947 min	0.942 min	2.31 min
pLSA	20 Newsgroups	1296 min	24.3 min	0.927 min	0.926 min	2.347 min
pLSA	French dataset	1109 min	31 min	2.5 min	2.47 min	6.01 min
GLDA	Lenta	81 min	0.9 min	0.042 min	0.08 min	3.39 min
GLDA	20 Newsgroups	281 min	3.78 min	0.123 min	0.197 min	11.153 min
GLDA	French dataset	2310 min	8.5 min	0.1 min	0.171 min	9.906 min
VLDA	Lenta	780 min	25 min	0.969 min	1.114 min	3.951 min
VLDA	20 Newsgroups	1320 min	40 min	2.933 min	3.035 min	10.69 min
VLDA	French dataset	2940 min	73 min	2.949 min	3.129 min	10.71 min

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
