# Peer review of "Renormalization Analysis of Topic Models"

_entropy, 2020, doi:10.3390/e22050556_

Round 1

Reviewer 1 Report

The paper has strong research materials.

I have a few technical comments in equations (1) and (7).

I have written my comments in red on the attached manuscript.

The equation (1) is not correct as it is presented. The p(w|t) should

be replaced by p(w|t,d). The probabilistic argument is

p(w|d) = ∑   p(w,t|d) = ∑   p(w|t,d) p(t|d). 

              t∈T                t∈T

Note that p(w,t|d) = p(w,t,d)/p(d) = [p(w,t,d)/p(t,d)][p(t,d)/p(d)]

                             = p(w|t,d) p(t|d).

This incorrectness has implication in the representations afterwards.

The equation (7) represents a mixture distribution.

It has to be mentioned in the paper.

Author Response

Dear reviewer,

thank you for your valuable remarks. Please, find point-by-point replies to your comments.

  1. The equation (1) is not correct as it is presented.   

Basic probabilistic topic models which we consider in our paper, exploit a conditional independence assumption that document d and word w are independent conditioned on the latent topic t, i.e. words,  conditioned on the latent class t, are generated independently of the specific document identity d [Hofmann “Unsupervised Learning by Probabilistic Latent Semantic Analysis”]. Therefore, p(w|d,t)=p(w|t).
We inserted a comment on the conditional independence assumption of topic models (lines 99-102).

  1. The equation (7) represents a mixture distribution.

The equation (7) represents a linear combination of probability density functions (in particular, probability mass functions), however, it can not be considered directly as a mixture distribution since it does not sum up to 1. But, after normalization, it does sum up to one and is, indeed, a probability distribution. 

The idea behind equation (7) is similar to applications of mixture distributions to model a statistical population with subpopulations, considering subpopulations as smaller topics to be merged.

We inserted a comment on equation (7) in the text of our article (lines 256-259).

Reviewer 2 Report

The main goal of the paper to find the optimal number of topics considering space and time complexity. While some novel approaches have been proposed, this paper is incomplete at the moment for the following reasons and it would be hard to give more detailed reviews on this paper now. Some conclusions could only be made after comparisons and references are made to the important missing works.

  1. It is surprising that the authors have not even cited works on Bayesian non-parametric approaches to topic modelling. Models such as Hierarchical Dirichlet Processes are completely missing from the paper.
  2. I was wondering what the final goal of the paper - speeding up the models, or finding an optimal number of topics? This should be clarified further. In the first two pages, it seems that as a reader, I am completely lost.
  3. The authors should also consider doing some more quantitative results such as document classification, document retrieval to convince a reader that the proposed new inference methods do not lead to inferior results when compared with traditional inference techniques. Proper tuning strategy should be considered for the baseline models. Several topic modelling papers have considered such experiments. I would request the authors to do some background study on these experiments.

Author Response

Dear reviewer,

thank you for your valuable remarks. Please, find point-by-point replies to your comments.

  1. “No citations on Bayesian non-parametric models”.

We inserted a discussion of non-parametric models (HDP models) and mentioned their limitations (lines 116-138).  In work [“Analyzing the Influence of Hyper-parameters and Regularizers of Topic Modeling in Terms of Renyi Entropy”, Entropy, 2020] we demonstrated that two particular implementations of HDP model did not infer the true number of topics on the datasets with the known number of topics. Moreover, we implemented numerical experiments with a hierarchical non-parametric model, namely,  hLDA [“The Nested Chinese Restaurant Process and Bayesian Nonparametric Inference of Topic Hierarchies”, Blei, Girffiths and Jordan] and found out that the output (in particular, the inferred numbers of topics) depends significantly on the values of hyper-parameters and differs for several runs of the model on the same dataset and with the same hyper-parameters (https://github.com/hse-scila/hlda-tests). These drawbacks are usually incompatible with reliability requirements set by TM end users . The study of such Nonparametric models requires a separate work and a separate article. But you can see our preliminary results. You can found out our simulation and python scripts about simulation of HPAM and HLDA and calculation of Renyi entropy as function of hyper-parameters.

  1. “The final goal of the paper in unclear”.

Our article is devoted to the application of the renormalization procedure for selecting the number of topics. Our approach combines a developed earlier Renyi entropy approach and renormalization technique. Thus, our approach allows us to speed up significantly the process of searching the optimal number of topics in comparison to standard grid search. We inserted clarifications in the introduction and highlighted the aim of our work (lines 38-52).

  1. “More quantitative results are needed to test new inference method”. 

In our article, we do not propose a new inference algorithm for topic models, but we propose a new approach for fast tuning of the existing topic models. In works [“Estimating Topic Modeling Performance with Sharma–Mittal Entropy”, Entropy, 2019. “Analyzing the Influence of Hyper-parameters and Regularizers of Topic Modeling in Terms of Renyi Entropy”, Entropy, 2020] we compared classical techniques for selecting the number of topics (such as log-likelihood, perplexity, semantic coherence) with Renyi entropy approach for the baseline topic models and found out that Renyi entropy approach leads to the best results in terms of accuracy. Therefore, in this paper, we focus on Renyi entropy approach and aim to speed it up. We inserted comments on the aim of our work and on the proposed method in order to avoid misunderstanding (lines 38-52). Moreover, we inserted comments on the comparison of Renyi entropy approach and the standard methods for selecting the number of topics (lines 31-34). We would like to highlight that our approach can not be used as a new inference method, therefore, it seems that it should not be tested in the proposed manner. However, we considered an additional dataset to test our approach and presented the results in subsection 3.9.

Reviewer 3 Report

The authors propose an approach to determining the number of topics in topic modeling, that consists in exploiting renormalization in conjunction with the Renyi entropy. Renormalization is a procedure, that involves merging pairs of topics. Three distinct approaches for choosing the pairs of topics to be merged are investigated. With such a contribution, the authors also aim to speed-up a previous entropy-based approach [13], in addition to formalize and extends the ideas in two earlier works [15,16].

The presented contribution is interesting. It is fluently-readable and clearly understandable. The empirical results are also interesting, in addition to proving the validity of the proposed approach.

Despite the above pros, the authors’ contribution requires some improvements.

Section 1 must explicitly summarize the contributions of the paper. In particular, those contributions, that are specifically devised to extend the proposed contribution with respect to authors’ previous research have to be specified in detail. This would allow the reader to immediately appreciate the novelty of authors’ additional effort. Moreover, Section 1 does not motivate the proposed approach against an elegant advance in Bayesian nonparametric modeling, i.e., coupling the Hierarchical Dirichlet Process (HDP) with a topic model (such as, e.g., in the case of HDP-LDA), which allows for avoiding the specification of the number of topics in advance. For the sake of completeness, Section 1 has to motivate the proposed approach against the foresaid HDP-based topic modeling.

Lines 57 – 63 should be removed from Section 1 and placed into an ad-hoc (currently missing) subsection of Section 2.

Overall, the current Section 2 may be reduced to contain all preliminaries (including lines 57 – 63) as well as Subsection 2.1 and Subsection 2.2. Being the core of the proposed contribution, Subsection 2.3, Subsection 2.4, Subsection 2.5 and Subsection 2.6 may be grouped to form a new Section 3. The newly resulting Section 2 may be enriched with a survey of topic models. In the name of self-containment, among the others, the latter survey should detail pLSA, LDA and GLDA.

Equation (7) should be further commented.

Lines 249 – 255 may be moved inside the current Section 3., as the description of the experimental purposes.

In Section 3.1, only two data sets are used for the experiments. Moreover, as emphasized both in the Abstract and in Subsection 3.5, the authors’ contribution addresses topic modeling in the domain of big data and, from this perspective, the two chosen data sets are relatively small. A larger number of bigger data sets is useful to provide a more solid and thorough understanding of the validity of the proposed approach.

Author Response

Dear reviewer,

thank you for your valuable remarks. Please, find point-by-point replies to your comments.

  1. “Summarization of the contributions is needed in section 1”.

We inserted the goals of this work and discussed the advantages of the renormalization approach with respect to the approaches in our previous works (lines 38-52). 

  1. “No discussion of nonparametric models”.

We inserted a discussion of non-parametric models (HDP models) and mentioned their limitations (lines 116-138).  In work [“Analyzing the Influence of Hyper-parameters and Regularizers of Topic Modeling in Terms of Renyi Entropy”, Entropy, 2020] we demonstrated that two particular implementations of HDP model did not infer the true number of topics on the datasets with the known number of topics. Moreover, we implemented numerical experiments with a hierarchical non-parametric model, namely,  hLDA [“The Nested Chinese Restaurant Process and Bayesian Nonparametric Inference of Topic Hierarchies”, Blei, Girffiths and Jordan] and found out that the output (in particular, the inferred numbers of topics) depends significantly on the values of hyper-parameters and differs for several runs of the model on the same dataset and with the same hyper-parameters (https://github.com/hse-scila/hlda-tests). These drawbacks are usually incompatible with reliability requirements set by TM end users.  The study of such Nonparametric models requires a separate work and a separate article. But you can see our preliminary results. You can found out our simulation and python scripts about simulation of HPAM and HLDA and calculation of Renyi entropy as function of hyper-parameters.

  1. “Re-structuring the article is necessary”.

The former lines 57-73 are removed from introduction and placed into subsection 2.1 (lines 96-115).  The former subsections 2.3-2.6 are removed from section 2 and placed into section 3  (lines 226-316). 

  1. “Describe pLSA, LDA and GLDA”.

We inserted Appendix A with a short description of the used topic models, namely, pLSA, LDA and GLDA (lines 537-609).

  1. Equation (7) should be further commented.

We inserted a comment on equation (7) (lines 256-259).

  1. “Experiments with a larger number of bigger datasets would be useful”.

We consider another bigger dataset in French language and inserted the obtained results in subsection 3.9 (figures 16, 17).

Round 2

Reviewer 3 Report

My concerns were properly addressed.

I would suggest the authors to rectify some typos such as, e.g., those reported below:

  • can not -> cannot;
  • principle -> principle;
  • a fast selecting the number of topics -> a fast selection of the number of topics;
  • menas -> means;
  • the comment at lines 601-603 should be checked.